# Expanding Insights: Harnessing Expansion Microscopy for Super-Resolution Analysis of HIV-1–Cell Interactions

**DOI:** 10.3390/v16101610

**Published:** 2024-10-15

**Authors:** Annett Petrich, Gyu Min Hwang, Laetitia La Rocca, Mariam Hassan, Maria Anders-Össwein, Vera Sonntag-Buck, Anke-Mareil Heuser, Vibor Laketa, Barbara Müller, Hans-Georg Kräusslich, Severina Klaus

**Affiliations:** 1Department of Infectious Diseases, Virology, Heidelberg University, 69120 Heidelberg, Germany; 2German Center for Infection Research (DZIF), Partner Site Heidelberg, 69120 Heidelberg, Germany

**Keywords:** expansion microscopy, ultrastructure expansion microscopy, super-resolution microscopy, HIV-1, HIV-1 post-entry, HIV-1 nuclear import, HIV-1 capsid, virus–host interaction

## Abstract

Expansion microscopy has recently emerged as an alternative technique for achieving high-resolution imaging of biological structures. Improvements in resolution are achieved by physically expanding samples through embedding in a swellable hydrogel before microscopy. However, expansion microscopy has been rarely used in the field of virology. Here, we evaluate and characterize the ultrastructure expansion microscopy (U-ExM) protocol, which facilitates approximately four-fold sample expansion, enabling the visualization of different post-entry stages of the HIV-1 life cycle, focusing on nuclear events. Our findings demonstrate that U-ExM provides robust sample expansion and preservation across different cell types, including cell-culture-adapted and primary CD4+ T-cells as well as monocyte-derived macrophages, which are known HIV-1 reservoirs. Notably, cellular targets such as nuclear bodies and the chromatin landscape remain well preserved after expansion, allowing for detailed investigation of HIV-1–cell interactions at high resolution. Our data indicate that morphologically distinct HIV-1 capsid assemblies can be differentiated within the nuclei of infected cells and that U-ExM enables detection of targets that are masked in commonly used immunofluorescence protocols. In conclusion, we advocate for U-ExM as a valuable new tool for studying virus–host interactions with enhanced spatial resolution.

## 1. Introduction

Fluorescence microscopy has been a cornerstone of biological and medical research since its conception over a century ago. However, many structures of interest, particularly in the field of virology, cannot be resolved using conventional fluorescence microscopy due to the diffraction limit of light microscopy (i.e., ~250 nm laterally and 500 nm axially). For instance, the human immunodeficiency virus type 1 (HIV-1) has a size of ca. 100–150 nm, which limits investigation of subviral structures and virus–host interactions using fluorescence microscopy [1].

To overcome this resolution barrier, several advanced microscopy techniques have been developed [2]. These methods were readily embraced by the virus research community, leading to critical insights into, e.g., virus morphology, assembly, and release [3,4,5]. However, most super-resolution techniques rely on expensive equipment and specific fluorescent dyes, are technically challenging, and demand substantial expertise in instrument operation, sample handling, as well as data analysis [6]. As a result, their widespread application remains limited.

Complementing these existing approaches, an alternative category of resolution-enhancing techniques known as expansion microscopy (ExM) has recently gained prominence [7]. Unlike other approaches that aim to improve resolution through modifications of optical elements or single-molecule localization microscopy (SMLM), ExM operates on the principle of isotropically expanding the sample in all dimensions. This expansion increases the distance between epitopes, allowing the visualization of sub-diffraction targets using conventional imaging techniques [7,8,9]. Sample expansion is achieved by the attachment of gel-linkable anchors to biomolecules within fixed and crosslinked samples before embedding the samples in a polycharged hydrogel [9]. To ensure uniform expansion, the samples subsequently undergo homogenization, which can be accomplished through proteolytic digestion or a combination of heat and detergent treatment [9]. Finally, incubation of the hydrogel in deionized water leads to sample expansion due to the repulsion of the charges within the hydrogel [9].

Following an initial proof-of-concept study by Boyden and co-workers [7], which demonstrated an approximately four-fold sample expansion in brain tissue slices and mammalian cells, ExM was rapidly adapted and optimized for different applications, resulting in a wide range of protocols tailored to meet different research needs (a detailed review of ExM principles and techniques has been provided by Wen 2023 and more recently by Hümpfer 2024 [8,9]). While some ExM adaptations have focused on retaining specific biomolecules such as lipids or RNA, others have aimed to optimize sample preservation or to increase the expansion factor, resulting in protocols capable of achieving sample expansion of up to 20-fold for a variety of targets [10,11,12].

A critical aspect of ExM is the visualization of structures of interest within the gel. Two principal strategies have emerged for visualizing protein targets: one involves staining samples prior to gel embedding and expansion [13,14,15], while the other entails staining targets after expansion within the gel [16,17,18]. Pre-expansion staining protocols typically utilize proteolytic digestion for homogenization, which destroys most epitopes and renders them unrecognizable for antibody labeling [9]. To overcome this limitation, proteins and other targets are often labeled via immunostaining before the expansion procedure, as proteolytic digestion generally allows retention of fluorescence [9]. While indirect antibody staining provides an easy and accessible method for localization of targets, it also introduces a linkage error of ca. 30 nm due to the size of the two antibodies necessary for visualization. To reduce this linkage error, the use of directly conjugated antibodies, which are commercially available, is a viable alternative. These antibodies can reduce the linkage error to around 15 nm compared to traditional indirect staining methods. Employing directly conjugated Fab fragments or nanobodies may further enhance localization precision, achieving a linkage error of approximately 3 nm. However, as with direct immunofluorescence with conventional antibodies, these approaches show a lack of signal amplification. In pre-expansion staining protocols, antibodies are subsequently expanded alongside targets, resulting in relative linkage errors akin to those observed in conventional microscopy [17]. Additionally, gel-embedding and proteolytic digestion can diminish the brightness of fluorophores [9].

In contrast, post-expansion labeling approaches utilize a gentler homogenization process via heat denaturation in the presence of detergent, which more effectively preserves sample epitopes compared to proteolytic digestion but may impair the fluorescence of dyes introduced before expansion [16,17,18]. This approach results in improved labeling efficiency due to the enhanced antibody access to target epitopes after expansion [17,19]. In addition, as fluorophores are generally introduced after expansion, fluorescence is not affected by the sample preparation. Post-expansion labeling may also reveal previously masked epitopes that become accessible during the expansion process [19]. Expanded samples can be imaged using various microscopy techniques, facilitating super-resolution imaging with classical fluorescence microscopy. By further combining post-expansion labeling strategies with advanced super-resolution techniques such as Airyscan or stimulated emission depletion (STED) microscopy [20,21], it is possible to increase the spatial resolution even further, approaching molecular resolution (<10 nm) and effectively bridging the gap between light and electron microscopy [17,19].

Despite the potential of ExM, it has not yet been extensively applied in virology, and, in particular, not in the field of HIV-1 research. In this study, we present an ExM strategy tailored for the analysis of virus-infected cells with a focus on subviral replication complexes in the nucleus. The nucleus serves as a key compartment for the replication of many viruses, which rely on the nuclear machinery for viral genome replication and gene expression [22,23]. Among the viruses that depend on the nuclear environment for replication is the retrovirus HIV-1 [24]. A defining characteristic of retroviruses is the integration of a reverse-transcribed double-stranded DNA copy of their genome into the host cell DNA. Since HIV-1 can infect non-dividing cells, it necessitates the import of the viral genome through the intact nuclear envelope [25]. After release into the cytosol of a newly infected target cell, capsids (also called viral cores) are transported towards the nucleus along the microtubule network [26,27,28]. Upon reaching the nucleus, capsids interact with cytoplasmic components of the nuclear pore complex (NPC) and traverse the NPC channel, with seemingly intact capsids being frequently observed in the nucleoplasm of infected cells [29,30,31,32]. Within the nucleus, capsids are eventually directed to nuclear speckles, where they accumulate and undergo genome uncoating, releasing the fully reverse-transcribed viral DNA for subsequent integration into host cell chromatin [30,32,33,34]. While recent studies shed light on the mechanisms involved in nuclear import of capsids [29,35], the molecular events involved in the targeting of nuclear speckles and in genome uncoating remain poorly understood.

Here, we establish and employ a post-expansion labeling ultrastructure expansion microscopy (U-ExM) protocol to visualize intra-nuclear structures involved in post-entry events during HIV-1 infection, applicable to both cell culture-adapted cell lines and primary immune cells [18,36]. We characterize the robustness of this protocol across different cell types and nuclear compartments, provide references for evaluating sample integrity, and emphasize the advantages of using a post-expansion labeling approach to probe viral structures.

## 2. Materials and Methods

### 2.1. Cell Culture and Virus Preparation

Cells were kept in a humidified incubator at 37 °C in a 5% CO_2_ atmosphere. HeLa-based TZM-bl cells (NIH AIDS Reagent program #8129-442, also called JC57BL-13) [37,38], and human embryonic kidney cells from the 293T/17 line (here referred to as HEK293T, #CRL-11268TM, purchased from ATCC, Kielpin Lomianki, Poland [39]) were cultured in Glutamax-I Dulbecco’s Modified Eagle’s Medium (DMEM) (Gibco-Thermo Fischer Scientific, Waltham, MA, USA) supplemented with 10% heat-inactivated fetal bovine serum (FBS, Capricon Scientific, Ebsdorfergrund, Germany), 100 U/mL penicillin, and 100 mg/mL streptomycin (Gibco-Thermo Fischer Scientific, Waltham, MA, USA). U2OS cells endogenously expressing Nup96-SNAP (CLS GmbH, St. Ingbert, Germany, #300444, clone 33 [40]) were cultured in HEK293T growth medium supplemented with 1× MEM NEAA (Gibco-Thermo Fisher Scientific, Waltham, MA, USA). Cell lines were regularly tested for mycoplasma contamination (MycoAlert^®^ mycoplasma detection kit, Lonza Rockland, Basel, Switzerland) and authenticated by STR profiling (Promega PowerPlex 21 Kit; carried out by Eurofins Genomics, Ebersberg, Germany).

Monocyte-derived macrophages (MDMs) were prepared from isolated buffy coats of healthy anonymous human blood donors (purchased from the Heidelberg University Hospital Blood Bank according to the regulations of the local ethics committee, S-023/2022 and S-025/2022, within the SFB 1129 [integrative analysis of the replication and spread of pathogens, funding period 07/2022–06/2026]) by Ficoll density gradient centrifugation using SepMate tubes (StemCell Technologies, Vancouver, BC, Canada). These human peripheral blood mononuclear cells (PBMCs) were cultured in GlutaMax-I Roswell Park Memorial Institute medium 1640 (RPMI 1640, Gibco-ThermoFisher Scientific, Waltham, MA, USA), supplemented with 10% heat-inactivated FBS, 100 U/mL penicillin, 100 mg/mL streptomycin, and 20 mM HEPES. After incubating for 3 h at 37 °C in a 5% CO_2_ atmosphere, non-adherent cells were discarded, and the adherent PBMCs were washed with fresh medium. These adherent PBMCs were then differentiated by maintenance in activation medium (GlutaMax-I RPMI 1640 containing 10% heat-inactivated FBS, 100 U/mL penicillin, 100 mg/mL streptomycin, 20 mM HEPES, and 10% human AB serum (Sigma Aldrich, St. Louis, MO, USA)) for 7 days, allowing for their transformation into macrophages. The resulting MDMs were subsequently cultured in the activation medium for up to four weeks, with medium changes occurring every 3 days, as previously described [41].

CD4+ T-cells were isolated from buffy coats obtained from healthy blood donors (DRK-Blutspendedienst, Mannheim, Germany) with approval by the local ethics committee (S-023/2022 and S-025/2022). Negative CD4+ T-cell isolation was performed using the RosetteSep™ Human CD4+ T-Cell Enrichment Cocktail (STEMCELL Technologies Inc., Vancouver, BC, Canada) following the manufacturer’s instructions. Briefly, the buffy coats were mixed with the enrichment cocktail and incubated before being layered onto Ficoll-Paque^TM^ PLUS (GE Healthcare Bio-Sciences, Uppsala, Sweden) and centrifuged. After isolation, the cells were washed, treated with ammonium–chloride–potassium lysis buffer to remove red blood cells, and resuspended in complete T-cell medium consisting of GlutaMax-I RPMI 1640 supplemented with 10% fetal bovine serum (Capricon Scientific, Ebsdorfergrund, Germany), 100 U/mL penicillin/100 mg/mL streptomycin (Gibco-ThermoFisher Scientific, Waltham, MA, USA), and 10 ng/mL IL-2 (Biomol GmbH, Hamburg, Germany). The cells were left in a resting state or activated using Dynabeads^®^ Human T-Activator CD3/CD28 (Gibco-ThermoFisher Scientific, Waltham, MA, USA) at a bead-to-cell ratio of 1:5 and cultured in 24-well plates at a cell density between 1 and 2 × 10^6^ cells/mL for 72 h in a humidified incubator at 37 °C with 5% CO_2_.

For production of non-infectious, RT-competent HIV-1 (NNHIV) particles, HEK293T cells were seeded into 10 cm dishes at a density of 5 × 10^6^ cells in 6 mL per dish, one day prior to transfection. The following day, the cells were transfected using a standard calcium phosphate procedure with a total of 10 µg of DNA per 10 cm dish. The transfection mixture included the non-infectious proviral plasmid NNHIV env(stop) ANCH [29,30], a Vpr.IN.eGFP expression plasmid containing the corresponding mutations in the IN gene on the NNHIV plasmid [42], and a pCMV-VSV-G expression plasmid (Addgene plasmid #8454, a gift from Bob Weinberg [43]) in the ratio of 7.7:1.3:1.0 µg. After 4–6 h, medium was replaced, and the cells were incubated at 37 °C for an additional two days prior to harvesting the supernatant. The supernatant was then centrifuged at 300× *g* for 5 min to remove cell debris and subsequently filtered through 0.45 µm mixed cellulose ester (MCE) filters. The filtered supernatant was layered onto a 20% (*w*/*v*) sucrose cushion and centrifuged at 107,000× *g* for 1.5 h at 4 °C. The resulting viral particles were resuspended in 30 µL of ice-cooled phosphate-buffered saline (PBS) containing 10% FBS and 10 mM HEPES (pH 7.5), pooled, rapidly frozen in liquid nitrogen, and stored at −80 °C. Viral stocks were quantified using the SYBR Green-based Product Enhanced Reverse Transcription assay (SG-PERT) as described previously [44].

### 2.2. Cell Seeding and Fixation

MDM and TZM-bl cells were directly seeded onto 6–18 mm glass coverslips, which were placed in multi-well plates, at least 24 h before fixation. Cell density was adjusted to reach 70–90% confluency on the day of fixation. CD4+ T-cells were seeded onto polyethylenimine (PEI)-coated 12 mm coverslips at a density of 300,000 cells per coverslip and incubated for 40 min at 37 °C in a 5% FBS/PBS solution. The PEI coating was applied for 1 h at 37 °C using 100 µL of a 0.5 mg/mL PEI solution. Prior to transferring the cells, the coated coverslips were washed three times with ddH_2_O. For fixation, the medium was removed, and the coverslips were washed once with PBS. Fixation was performed for 10 min at room temperature using 4% paraformaldehyde (PFA, Electron Microscopy Sciences, Hatfield, PA, USA, #15719)/0.0075% glutaraldehyde (GA, Sigma Aldrich, Taufkirchen, Germany, G5882-10X1ML) in PBS. Subsequently, the coverslips were washed three times with PBS before either proceeding to U-ExM or being stored in PBS at 4 °C. For imaging during the U-ExM process, nuclei were briefly stained with 1 mg/mL Hoechst 33342 (here referred to only as Hoechst, Thermo Fisher Scientific, Waltham, MA, USA) for 5 min at room temperature in the dark. Afterwards, the coverslips were washed twice with PBS before imaging and subsequent expansion.

For determination of optimal fixation conditions for maintaining cytoplasmic integrity, cells were washed and fixed using either a pH 6.8 PEM buffer (composed of 80 mM PIPES, 5 mM EGTA, and 2 mM MgCl_2_ in ddH_2_O) as described by Leyton–Puig [45] or a pH 7.4 PHEM buffer (containing 60 mM PIPES, 25 mM HEPES, 10 mM EGTA, and 2 mM MgCl_2_ in ddH_2_O) as outlined by Sobue [46] instead of PBS. For fixation with a pH 6.1 cytoskeleton buffer (10 mM MES, 150 mM NaCl, 5 mM EGTA, 5 mM glucose, and 5 mM MgCl_2_ in ddH_2_O) following the protocol of Small [47], cells were rinsed once with pre-warmed cytoskeleton buffer. Cells were then detergent extracted with 0.25% Triton X-100/0.3% GA in cytoskeleton buffer for 90 sec at room temperature, followed by fixation with 2% GA in cytoskeleton buffer for 15 min at room temperature [45,48,49]. After fixation, cells were rinsed once with cytoskeleton buffer and then quenched with 50 mM NH_4_Cl in PBS for 10 min at room temperature. Finally, cells were washed twice with PBS before progression to U-ExM as described below.

For cryofixation, samples were treated according to the protocol provided by Laporte et al. [50]. Briefly, cells seeded on 6 mm coverslips were incubated briefly with either HEK293T growth medium or HEPES-buffered Hanks Balanced Salt Solution (HHBSS composed of 20 mM HEPES, 137 mM NaCl, 0.49 mM MgCl_2_, 1.26 mM CaCl_2_, 5.4 mM KCl, 0.41 MgSO_4_, 0.64 mM KH_2_PO_4_, 3 mM NaHCO_3_, and 5.5 mM glucose) containing 10% glycerol at pH 7.4. The samples were then plunge-frozen in liquid ethane that had been precooled at liquid nitrogen temperature using a Leica EM GP2 plunge freezer (Leica Microsystems, Wetzlar, Germany). The plunge freezer was operated as indicated by the manufacturer in a sample chamber environment of 25 °C and 80% humidity, a liquid ethane temperature of −183 °C, and using one-sided blotting on Whatman grade 1 filter paper (Sigma Aldrich, St. Louis, MO, USA; #1001-055) with a blotting time of 5 s. Subsequently, the samples were incubated in dry ice-chilled acetone (Sigma Aldrich, Taufkirchen, Germany, #270725-1L, HPLC, ≥99.9%), progressively transitioned to 0 °C before rehydration in a series of ethanol (Fischer Scientific-Thermo Fischer Scientific, Waltham, MA, USA, #E/0650DF/15 1L, ≥99.8%)/water baths with increasing water percentages [50]. Once the samples reached PBS incubation, samples were expanded as described below.

### 2.3. Ultrastructure Expansion Microscopy (U-ExM)

The protocol established by Gambarotto et al. [36] was implemented with minor modifications to optimize gel handling and reagent usage. A comprehensive protocol, including detailed information on all reagents, can be found in Appendix A. As previously described, sodium acrylate quality is critical for expansion. Here, we used sodium acrylate powder from AK Scientific (Union City, CA, USA; R624-5g), which was stored at −20 °C and consistently yielded good results in our hands.

The U-ExM protocol involves several steps (Table 1), each with extended incubation times. The approximate durations for each step are as follows: anchoring/crosslinking prevention (3.5 h), gelation (1.5 h), denaturation (2 h), first expansion (either 3× 30 min, or 2× 30 min and once overnight), gel shrinking (30 min), blocking (30 min), primary antibody incubation (2.5 h to overnight), washing (50 min), secondary antibody incubation (2.5 h), washing (50 min), and final expansion and staining with synthetic fluorescent sphingosine BODIPY-Ceramide dyes (BODIPY, Thermo Fisher Scientific, Waltham, MA, USA) staining (1× 30 min, overnight, and 2× 30 min). In some cases, we included an additional staining step using a fluorescently conjugated NHS-ester (ATTO-TEC, Siegen, Germany) after the second antibody incubation and washing, which extended the protocol for an additional 2 h to account for incubation and washing. To mitigate contamination risks during the long incubation times, particularly those involving bovine serum albumin (BSA, fatty acid free from Carl Roth^®^, Karlsruhe, Germany), we supplemented all solutions with 0.01% NaN_3_ (*w*/*v*) for any incubation lasting 30 min or longer.

Although the U-ExM protocol can be completed over three days, we opted for a two-day schedule for the expansion process, with imaging typically conducted on the third and fourth days.

#### 2.3.1. Sample Mounting

Gel pieces were imaged using either Poly-D-Lysine-coated ibidi 8-well glass bottom µ-slides (ibidi GmbH, Gräfeling, Germany) or uncoated 6-well plates (Corning GmbH, Kaiserslautern, Germany). To prepare the ibidi-coated µ-slides, 200–500 µL of a 0.1 mg/mL. Poly-D-Lysine solution (Gibco-Thermo Fisher Scientific, Waltham, MA, USA) was added to each well and incubated for 1 h at 37 °C. Afterwards, the wells were washed three times with ddH_2_O. The orientation of cells within the gels was determined using a Nikon eclipse Ts2 widefield inverted benchtop microscope (Nikon Instruments Inc., Tokyo, Japan) by examining either Hoechst or BODIPY staining through a 10×/0.25 air objective. For gel mounting, a piece of gel that fit into a well of an ibidi 8-well µ-slide was carefully excised and gently pressed, cell-side down, into the coated well to ensure leveling and minimize air bubbles. To prevent the gels from drying during imaging, 2–3 drops of ddH_2_O were added to the well.

Unmounted, expanded, and stained gel pieces were kept in ddH_2_O supplemented with 0.01% NaN_3_ for up to one week, allowing for later mounting and imaging.

#### 2.3.2. Expansion Factor Measurement

Since only sections of the original gel were used for imaging, it was not possible to determine the final expansion factor by measuring the final gel diameter, as has been carried out previously [36]. Instead, we measured the diameter of the gel after shrinkage and just prior to excising the gel pieces to determine an initial expansion factor, EF_PBS_. Rectangular gel pieces were then excised and measured before proceeding with staining and expansion. After the final expansion and prior to mounting, we measured the fully expanded gel piece again to determine a second expansion factor, EF_H2O_. Consequently, the final expansion factor EF_Final_ was calculated as follows:EF_Final_ = EF_PBS_ × EF_H2O_(1)

### 2.4. Fluorescence Microscopy

Imaging of samples during the U-ExM process was conducted using a Nikon eclipse Ts2 widefield inverted benchtop microscope (Nikon, Tokyo, Japan) equipped with a 10×/0.25 Ph1 ADL air objective (working distance (WD) 6.2 mm) and a 385/470/560 filter set. Images were acquired with a CMOS DMK 33UX174 camera (theimagingsource, Bremen, Germany) using the NIS-Elements D software (version 5.02, Nikon, Tokyo, Japan). Exposure time, gain, and gamma values were optimized to enhance contrast and visibility.

For large overviews of gel pieces as well as GelMap coverslips (Utrecht University, Utrecht, Netherlands) [51], we employed the Zeiss CellDiscoverer 7 automated microscope (Zeiss, Oberkochen, Germany). Gels and coverslips were imaged in either 6-well plates or ibidi 8-well glass bottom µ-slides. Images were acquired using either a Plan-Apochromat 5×/0.35/WD 5.1 mm or a Plan-Apochromat 20×/0.7/WD 2.2 mm autocorr air objective with 1×–2× tube lens magnification and a Zeiss Axiocam 712 sCMOS camera for detection. Multichannel images were acquired using light-emitting diodes at 385 nm, 470 nm, 567 nm, and 625 nm for excitation, paired with a quad-bandpass filter (QBP 425/30 + 514/30 + 592/25 + 709/100 nm) for emission. Image tiling was executed with a 10% overlap using the *Definite Focus* module for focus stabilization.

Point laser scanning confocal microscopy was performed on a Zeiss LSM900 microscope (Zeiss, Oberkochen, Germany) equipped with the Airyscan 2 detector. Imaging was performed using either a Plan-Apochromat 20×/0.8/WD 0.55 mm air or a Plan-Apochromat 63×/1.4/WD 0.193 mm oil immersion objective. Similar to previous imaging, tiling was performed with 10% overlap and the *Definite Focus* module for focus stabilization. Multichannel images were acquired sequentially in either frame or stack scanning mode, employing diode lasers at 405 nm, 488 nm, 561 nm, and 640 nm for imaging of fluorophores in the blue, green, orange, and red spectra of the visible light, respectively. Emission detection was configured using variable dichroic mirrors, and detection was optimized for emission spectra of different fluorophores. GaAsP PMT or Airyscan detectors were ultilized, with gain typically set between 700 and 900 V, while the offset remained at 0%. Sampling was optimized for Nyquist criteria (ca. 50 nm laterally and 250 nm axially for the 63×/1.4 oil immersion objective), conducted bidirectionally at maximum scanning speed with 2× line averaging. Post-acquisition processing of 2D or 3D Airyscan images was performed using ZEN Blue (version 3.1, Zeiss, Oberkochen, Germany), applying the default Airyscan Filtering (AF) strength.

STED microscopy was performed on an Expert Line STED system (Abberior Instruments GmbH, Göttingen, Germany) equipped with an SLM-based *easy3D* module and an Olympus IX83 microscope body with a 100× oil immersion objective (NA 1.4; WD 0.13 mm; Olympus UPlanSApo). STED images were acquired using a 640 nm excitation laser and spectral detectors, with detection ranges optimized for fluorophore emission accordingly. A 775 nm STED laser (operating at 10–20% of the maximal power of 3 mW) was used for depletion, with a pixel dwell time of 10 μs and 15 nm xy sampling.

### 2.5. Image Selection, Processing, and Analysis

For general image visualization and analysis, the FIJI distribution of the open source software ImageJ (version 1.54f) was used [52]. Images were contrast adjusted to enhance visibility and filtered with a mean filter with a radius of 1–2 pixel. Data graphs were generated and statistical analysis performed using GraphPad Prism version 5.00 for Windows (GraphPad Software, San Diego, CA, USA). Images presented were selected as representative examples of 2–3 biological replicates, with at least 3 images (single planes or z-stacks) being acquired per replicate after visual examination of samples. For samples infected with NNHIV, infected cells were identified by the presence of CA- and high CPSF6-containing structures in the nucleus before acquisition of 5–7 images over 2 biological replicates.

#### 2.5.1. Stitching

For tiled images, stitching was performed using Zen Blue version 3.9 without shading correction, using the nuclear channel as a reference.

#### 2.5.2. Bleedthrough Correction

Due to the usage of antibodies and dyes with spectral overlap as well as strong fluorescence intensity in some channels, some bleedthrough was observed in multi-channel imaging during point laser scanning confocal microscopy. To determine if fluorophore bleedthrough was present, images were examined for the presence of duplicate structures in multiple channels. If bleedthrough was present in a channel, a subtraction-based correction approach was used to locally remove bleedthrough in the affected image [53]. Subtraction-based bleedthrough correction can be used in cases where spectral crossover is only present in one direction, i.e., if there is a reference channel without any bleedthrough available, as was the case here.

First, the contribution of the fluorophore that causes bleedthrough was determined in the channel that was affected by bleedthrough. To do so, in the image that was affected by bleedthrough, the mean signal intensity was calculated in an area that only showed bleedthrough. Then, the mean signal intensity was calculated in the same area of the channel where bleedthrough originated. Subsequently, a bleedthrough correction factor was determined by dividing the mean signal intensity of the channel where bleedthrough originated by the mean signal intensity of the channel that was affected by bleedthrough. To create a correction image that represented the bleedthrough, the original channel where the bleedthrough originated was then divided by the calculated correction factor. The resulting lower intensity correction image was then subtracted from the channel that was affected by bleedthrough via the *Image Calculator* module in ImageJ (Appendix A).

#### 2.5.3. Nucleus Segmentation and Nuclear Diameter Measurements

To determine the dimensions of the nucleus before and after expansion, the same set of nuclei in both conditions were segmented. Prior to segmentation, images of nuclei acquired before expansion were processed using a median filter with a radius of 1 pixel. Subsequently, the nuclear channels of both images were segmented using the pre-trained cell segmentation algorithm CP within the cellpose 2.2.1 framework (platform Win32, python version 3.8.16, and torch version 1.12.0) [54]. Label images were exported as PNG files and further processed in ImageJ. To separate nuclei before analysis, an erosion in xy of 1 pixel was carried out using the ImageJ *3D Suite Close Labels* function [55]. The images were then thresholded, and the Feret diameter was measured using the *Particle Analyzer* function in ImageJ with no restrictions for circularity or size.

#### 2.5.4. NPC Dimension Measurements

For evaluating the properties of the Nup153 foci as a proxy for the dimensions of the NPC, a maximum intensity projection of 21 z-slices, each spaced 140 nm apart, starting from the bottom of a U2OS nucleus was created. The image was thresholded (600–65,535) to isolate Nup153 signal foci, and the resulting binary image was analyzed using the *Analyze Particles* function in ImageJ, excluding all objects at the borders of the image and only including objects with a size of 20 pixel^2^ and larger. Subsequently, the maximum and minimum Feret diameters were divided by the expansion factor EF_Final_ to account for expansion.

#### 2.5.5. Nup153 and Nup96-SNAP Signal Intensity Measurement

To measure the signal intensity of the Nup153- and Nup96-SNAP channels, a line was drawn through an area of the nuclear envelope containing several distinct Nup153 foci. Signal intensities along the line were measured in both channels using the *plot profile* function of ImageJ. The intensity values were exported and normalized to the minimum and maximal values for each channel. During plotting, the graphs were aligned to the point of highest intensity to compensate for any potential shifts that may have occurred during imaging.

## 3. Results

Expansion microscopy has gained significant traction within the broader research community; however, its application in virology remains limited, with particularly few studies exploring HIV-1-related phenomena [56,57]. Other studies have investigated general aspects of the broader virology field, such as the spatial distribution of virions and viral proteins in infected cells and tissues, as well as host cell alterations following infection [58,59,60,61]. Among the available expansion protocols, the U-ExM method, which offers a 4–5-fold expansion alongside a post-expansion labeling approach, has found broader application in fields such as parasitology, centriole biology, and neurobiology, where it has significantly improved the visualization of target structures [62,63,64,65,66]. Here, we aim to assess the effectiveness of the U-ExM protocol to probe processes during virus infection, with a particular focus on nuclear HIV-1–cell interactions at high resolution.

### 3.1. The Ultrastructure Expansion Microscopy Protocol Allows Robust Expansion with Good Preservation of Cellular Morphology

We first benchmarked the U-ExM protocol for application in TZM-bl cells, a HeLa-derived cell line frequently used in HIV-1 research [38]. To establish fundamental expansion parameters, we adhered closely to the protocol outlined by Gambarotto in 2021 [36], with minor modifications to optimize sample handling and reagent usage (refer to Materials and Methods as well as Appendix A for a detailed protocol). To monitor the expansion process and the cell state during the U-ExM protocol, we seeded TZM-bl cells onto glass coverslips and subjected them to the U-ExM process. We employed nuclear staining using Hoechst at low concentrations directly after fixation to visualize the position and morphology of nuclei. Imaging was conducted following fixation, after incubation with the anchoring solution, immediately after gelation within intact gels sandwiched between the sample coverslip and a rectangular glass slide, as well as in detached gels post-denaturation (Figure 1A,B). Cells were clearly visible at all time points prior to denaturation, exhibiting normal cellular and nuclear morphology and size (Figure 1B). Although the Hoechst signal was lost after denaturation, most likely due to the harsh conditions necessary for sample homogenization, cells remained discernible in transmitted light (Figure 1B). Consistent with the observed size increase of the gel during denaturation, longitudinal cell size increased from ca. 30 µm to up to 100 µm in the gels post-denaturation (Figure 1B). Following the U-ExM protocol, gels were fully expanded for the first time via incubation in deionized water overnight and subsequently shrunk again by incubation in PBS to facilitate sample handling. To minimize reagent usage for staining, we excised 1 cm × 0.5 cm pieces from the center of the gel for blocking and staining. For initial characterization of cellular and nuclear morphology, chromatin was stained using Hoechst, and samples were incubated with BODIPY during the final expansion to visualize lipid membranes, as previously established [67]. Measurement of gel and coverslip sizes before and after expansion indicated an expansion factor of ca. 4.3, in line with previous studies in mammalian cells and other eukaryotes [18,63,67]. After the final expansion, cells showed a normal overall morphology within the gel, with longitudinal cell sizes often exceeding 100 µm, consistent with the calculated expansion factor (Figure 1B). It is noteworthy that samples were more prone to degradation and contamination after expansion; hence, we consistently employed 0.01% sodium azide for any incubation or storage period exceeding one hour.

Next, we assessed whether samples experienced distortion during the expansion process by directly comparing cells before and after expansion. In traditional pre-expansion labeling protocols, structures can be imaged both prior to and following expansion, allowing direct comparisons. However, in the case of U-ExM, direct comparisons are more challenging, as staining typically occurs only after expansion. To identify the same set of cells before and after expansion as well as to visualize any sample deformation, we employed the GelMap system developed by the Kapitein lab [51]. The GelMap consists of a coverslip printed with a defined fluorescently labeled protein pattern comprising a 20 × 20 µm grid along with numerical and letter markers (Appendix A). For reference images, we acquired wide-field images of fixed TZM-bl cells seeded on an 18 mm GelMap coverslip patterned with a custom R2-myc-his nanobody conjugated to ATTO 647N (Figure 1C and Appendix A). Subsequently, the whole GelMap coverslip was expanded, and a 1 cm × 0.5 cm gel piece was excised from the center of the gel before blocking and staining. Although the ATTO 647N fluorescence was lost during the U-ExM procedure, the GelMap grid could be visualized via α-R2-myc immunofluorescence staining in wide-field microscopy after expansion, allowing for identification of positions that had been imaged before expansion (Figure 1D and Appendix A).

To compare the appearance of cells before and after expansion, we segmented the same set of nuclei using the cellpose 2 CP algorithm on images taken before and after expansion, measuring the maximum diameter of these nuclei (Appendix A). Comparison revealed a maximum diameter of ca. 17 µm before and 73 µm after expansion, indicating a nuclear expansion of about 4.3-fold. This finding aligns with our measurements of the expansion factor based on gel size, suggesting that the observed increase in gel size is directly translatable to microscopic structures (Figure 1E). Although the overall distribution of nuclei was retained and cells maintained their relative position, the masks of the segmented nuclei before and after expansion could not be overlaid using rigid transformation or scaling, indicating local distortion (Figure 1C,D and Appendix A). This was further confirmed by inspection of the protein grid, which displayed small local deformations in the grid lattice and large disruptions in certain areas (Appendix A). Nevertheless, the overall shape and morphology of individual cells appeared to be well conserved (Figure 1C,D, Appendix A).

While most cells showed no apparent abnormalities, expansion artifacts were visible in 5–20% of the cells (Figure 1F,G). These artifacts could be categorized into two groups: (i) cells that appeared to not have been permeabilized during the expansion process (Figure 1G), and (ii) cells exhibiting cytoplasmic rupture (Figure 1F). In severe instances, cytoplasmic rupture resulted in the displacement of visually intact nuclei from empty and occasionally collapsed cytoplasmic shells (Figure 1F). Non-permeabilized cells appeared as opaque, cell-shaped structures lacking staining and could be identified already after denaturation (Figure 1B,G). Although both phenotypes were observed across different samples, they never occurred simultaneously. The abnormal cells were visually distinct from properly expanded cells, allowing for their immediate exclusion from further analysis.

### 3.2. Optimization of Fixation and Imaging Conditions

Although nuclei showed no visible abnormalities even in severe cases of cell rupture, inspection of the cytoplasmic compartment through Airyscan microscopy occasionally revealed gaps in the BODIPY membrane staining. This finding indicates compromised cytoplasmic integrity (Appendix A). To optimize the preservation of the cytoplasmic compartment during expansion, we employed various fixation procedures, expanded TZM-bl cells, and performed staining using an α-tubulin antibody alongside a membrane marker (BODIPY) and Hoechst. It is noteworthy that microtubules are prone to fragmentation under inadequate fixation conditions [50,68], while membrane staining was utilized to assess overall compartment integrity. We compared several fixation conditions, including cryofixation [50] in different solutions and chemical fixation with 4% PFA/0.0075% GA using different buffer conditions optimized for cytoskeletal conservation (Table 2, Figure 2A and Appendix A).

Regular chemical fixation using a 4% PFA/0.0075% GA in PBS solution resulted in fragmented microtubules and diffuse membrane signals when observed with Airyscan microscopy. In contrast, alternative fixation conditions, including chemical fixation in PHEM buffer and cryofixation in HHBSS/10% glycerol, showed uninterrupted microtubules and distinct membrane networks (Figure 2A and Appendix A). Notably, these different fixation conditions did not alter the appearance of the nuclear compartment (Figure 2A and Appendix A).

Next, we evaluated the 3D sample structure by employing confocal microscopy and optical sectioning. We acquired 3D stacks of whole TZM-bl cells after expansion using Airyscan microscopy, using BODIPY and Hoechst staining to visualize the cellular and nuclear compartments (Figure 2B, Appendix A). Reconstruction of cells showed robust 3D expansion without apparent distortion of the cell body or the nucleus (Figure 2B, Appendix A). As expected for a 4.3-fold expansion, cells regularly reached heights of over 50 µm (Figure 2B), which resulted in extended imaging times at maximal resolution. In addition, we observed a decrease in fluorescence intensity when acquiring 3D volumes and imaging deeper into the sample (Figure 2B,C, Appendix A). This reduction in signal intensity was likely due to a combination of increased scattering and bleaching, which were particularly pronounced with fluorophores within the blue light spectrum, including Alex 405 Plus, Hoechst, and BODIPY (Figure 2B,C, Appendix A). Nevertheless, the signal degradation in general stains like Hoechst and BODIPY was largely negligible for the identification of relevant structures (Figure 2C). To prevent signal degradation during imaging of immunostained targets, we also employed fluorophores developed for STED microscopy, including long-stokes shift dyes, which exhibit greater resistance against bleaching [69].

### 3.3. U-ExM Achieves Robust Expansion of Different Cell Types

We subsequently investigated the applicability of the expansion procedure established for the TZM-bl model cell line across different cell types. Here, we focused on cell types relevant for HIV-1 research, particularly primary cells recognized as significant HIV-1 reservoirs [70]. To this end, we fixed and expanded both resting and activated primary CD4+ T-cells as well as monocyte-derived macrophages (MDMs). To evaluate the overall morphology of the expanded cells, we employed a combination of Hoechst and BODIPY staining. Additionally, we used immunofluorescence staining for the inner nuclear pore complex (NPC) protein Nup153 to investigate the NPC distribution and morphology. All cell types were robustly expanded, achieving a ca. 4.3-fold size increase while maintaining intact cell bodies (Figure 3A).

Airyscan microscopy of expanded cell nuclei revealed distinct chromatin domains, characterized by varying degrees of Hoechst staining, likely representing euchromatin (low intensity) and heterochromatin states (high intensity) (Figure 3A). Notably, strong Hoechst staining was prominent at the nuclear periphery and around nucleoli, which are known to serve as heterochromatin hubs [71]. Interestingly, while resting CD4+ T-cells exhibited a high density of condensed chromatin, activated T-cells showed a larger proportion of euchromatin (Figure 3A). This difference likely reflects the distinct chromatin architecture resulting from the varying transcriptional programming between the activated and resting states [72].

Expansion resulted in substantial improvement in rendering the cytoplasmic T-cell compartment, which is typically challenging to visualize by fluorescence microscopy (Figure 3A). All cells displayed a complex membranous network throughout the cytoplasm, with the nucleus containing markedly less BODIPY staining than the cytoplasm (Figure 3A), consistent with the lower lipid and membranous structure contents in the nucleus. In contrast, nucleoli showed a relatively high BODIPY intensity (Figure 3A).

Staining for Nup153 revealed numerous distinct spots decorating the nuclear envelope. The Nup153 foci had a diameter of around 400 nm (corresponding to ca. 100 nm in the pre-expansion state, as expected for nuclear pores [73]) and were primarily localized at the periphery of the chromatin staining, indicating that these foci represent individual NPCs (Figure 3A).

### 3.4. Different Cellular Structures Are Well Conserved after Expansion

We next focused on the preservation of the nuclear compartment by visualizing different, well-characterized nuclear structures. In non-diving cells, the nucleus acts as a closed compartment, presenting a potential barrier to invading viruses [22,23]. These viruses must hijack the cellular nuclear import machinery, or the NPC, to gain entry into the nucleus [22]. To assess whether U-ExM can be used to investigate virus–host interactions at the nuclear pore, we first probed the ultrastructure of the NPC and nuclear periphery in their unperturbed state. A structurally conserved and well-defined feature of the nuclear periphery is the presence of chromatin-free regions below the NPC, which has already been described in early studies on nuclear architecture using electron microscopy [71]. We fixed and expanded primary MDMs and visualized the different structures of the nuclear periphery by using BODPY to stain the nuclear envelope membrane, Hoechst to visualize the chromatin structure, and Nup153 immunostaining to mark the position of the NPC. Using Airyscan microscopy, we could clearly observe the individual layers of the nuclear envelope in some regions where it was oriented perpendicular to the imaging plane due to the enhanced resolution provided (Figure 3B). Additionally, even in regions where such distinct visualization of the nuclear envelope layers was not achievable, we consistently observed heterochromatin-free regions beneath the NPC, as indicated by Nup153 immunostaining (Figure 3C). These heterochromatin-free regions appeared as narrow, elongated, channel-like structures that eventually connected and merged with larger heterochromatin-free areas further from the nuclear envelope (Figure 3C). These observed structural features align with the widely accepted model of the chromatin landscape at the nuclear periphery, where the interchromatin space is characterized by chromatin-free channels below the NPCs, connecting to a larger, continuous network throughout the nucleus [71].

The interchromatin space of the nucleus also harbors several specialized compartments that may be exploited by viruses for their own purposes [23]. Among these, nuclear speckles are the most abundant. These chromatin-free, membraneless organelles function as storage sites for splicing components and play a crucial role in RNA processing [74]. Another prominent nuclear compartment is the nucleolus, which is primarily involved in ribosome biogenesis [74]. Both nuclear speckles and nucleoli are characterized by low chromatin abundance, high protein density, and formation through phase condensation [74]. They can be visually distinguished by their shape and size: nucleoli are larger and spherical, whereas nuclear speckles are more numerous, smaller, and less defined in structure. Additionally, both compartments can be distinguished by the localization of the nuclear speckle structural proteins SRRM2 and SON, which form the main scaffold of nuclear speckles but are absent in nucleoli [74]. To investigate whether the structural integrity of these compartments is preserved after U-ExM, we fixed and expanded TZM-bl cells, staining them for chromatin using Hoechst and for the two nuclear speckle proteins SRRM2 and SON. We also employed fluorescently conjugated *N*-Hydroxysuccinimide (NHS)-ester to visualize protein distribution within the nuclear compartment, as this reagent reacts with primary amines in proteins, providing a general staining for protein density [10,36]. Airyscan microscopy revealed the presence of multiple chromatin-free regions of different sizes within the nucleus (Figure 3D). As expected, these chromatin-free regions showed higher protein density; however, only the smaller regions colocalized with SON and SRRM2, confirming their identity as nuclear speckles (Figure 3D,E). This finding indicates that the structure and composition of phase-separated nuclear bodies, such as nuclear speckles and nucleoli, are well preserved in U-ExM.

### 3.5. Nanoscopic Targets Are Well Preserved during Expansion

Since virus–host interactions predominantly occur at a nanoscale, we aimed to investigate whether U-ExM alters the structure of well-defined large macromolecular complexes. The NPC is among the best characterized and structurally defined assemblies within eukaryotic cells. Containing more than 500 individual proteins, the NPC forms a channel with an eight-fold symmetry, consisting of three stacked rings. The diameters of the circular NPC structures range from 40 to 60 nm in the inner ring’s channel to 150 nm in the distal components of the nuclear ring [29,73]. The nuclear basket protein Nup153 is located on the nuclear side of the NPC. Although the structure of the nuclear basket remains poorly defined, single-molecule localization data indicate that it occupies an area with a diameter of ca. 100 nm below the main body of the NPC [73]. When we probed MDM with immunostaining using a primary and secondary antibody cascade against Nup153, single NPC could not be resolved in Airyscan microscopy but could be distinguished using STED microscopy (Figure 4A). In contrast, when applying U-ExM to MDM followed by immunostaining and Airyscan microscopy, we could clearly distinguish individual, circular Nup153 signals, representing NPCs (Figure 4A). The combination of U-ExM and Airyscan microscopy reaches similar resolutions as STED microscopy while also reducing linkage errors during immunostaining, thereby enhancing the separation of individual pore complexes (Figure 4A).

To further probe the morphology of the NPC, we subjected U2OS cells to U-ExM and performed Nup153 immunostaining alongside Hoechst staining to characterize the nanoscopic environment of the pore. We then acquired Nyquist-optimized 3D volumes of the nuclei close to the sample carrier using Airyscan microscopy (Figure 4B, Appendix A). In maximum intensity projections of 3D volumes, NPC appeared as distinct, round structures, again associating with heterochromatin-free regions (Figure 4B). To further explore the structure of the NPC after expansion, we measured the minimum and maximum diameter of the Nup153 signals in the expanded sample (Figure 4C and Appendix A). Considering the expansion factor of 4.3, we determined that the minimum and maximum diameters of Nup153 ranged from ca. 90–110 nm across a wide range of foci, with an aspect ratio of 1.2 between minimum and maximum diameter. These data indicate that nuclear pores were well conserved at the nanoscale (Figure 4C and Appendix A).

To push the limit of resolution that can be obtained with widely available imaging methods such as Airyscan microscopy, we stained expanded TZM-bl cells for the distal inner NPC component ELYS. This protein forms a ring structure composed of eight subunits, previously measured to have a diameter of ca. 150 nm using SMLM, making it one of the largest diameters of the NPC [73]. In maximum intensity projections of Nyquist-optimized 3D volumes at the bottom of nuclei stained using a primary and secondary antibody cascade, we successfully visualized some ring-like structures, although most ELYS signals appeared as punctuate structures (Figure 5A). This indicates that U-ExM, in combination with Airyscan microscopy, can resolve the subunit arrangement of structures with a total diameter of as small as 150 nm, even when employing common immunofluorescence strategies. We also probed the spatial separation of the distal cytoplasmic and nuclear components of the NPC after expansion, which are only separated by the length of the NPC itself. Using immunostaining against the peripheral cytoplasmic NPC proteins Nup88 and ELYS, we found a clear separation of both targets across the nuclear envelope in imaging planes perpendicular to the nuclear envelope when using Airyscan microscopy (Figure 5B).

Although the relative linkage error in U-ExM is reduced compared to both pre-expansion staining expansion workflows and conventional immunofluorescence analysis (ca. 7 nm compared to 30 nm), some uncertainty remains when probing nanoscopic structures. To further minimize linkage errors, several approaches have been developed, including the self-labeling protein tags such as SNAP-, CLIP-, or HaloTag. These tags bind directly and irreversibly to their substrates, resulting in a linkage error equivalent to the size of the tag itself (i.e., ca. 5 nm) [75]. Therefore, we aimed to evaluate the application of such tags in U-ExM using U2OS cells stably expressing the NPC protein Nup96 fused to the SNAP-tag [40]. While the fluorescence of SNAP substrates was lost when ligands were applied prior to expansion, we successfully visualized SNAP-tagged Nup96 by applying a biotin-coupled SNAP substrate to permeabilized cells before U-ExM, followed by detection using fluorescently labeled streptavidin after expansion (Figure 5C and Appendix A). Although streptavidin adds an additional 5 nm linkage error to the detection assembly, it still offers improved localization accuracy compared to antibodies [76]. The combination of biotin-coupled SNAP substrates and fluorescent streptavidin showed more unspecific fluorescence compared to indirect immunostaining against Nup153. However, nuclear envelope-associated Nup96-SNAP signals colocalized well with the Nup153 signal, indicating that this strategy allows the detection of any target that can be linked to biotin with low linkage error (Figure 5C,D and Appendix A).

### 3.6. U-ExM Can Be Used to Probe Different Steps of the HIV-1 Replication Cycle

Having explored sample preservation during U-ExM with a focus on HIV-1-related structures, we finally aimed to probe different steps in the post-entry phase of the HIV-1 replication cycle. To this end, we first investigated if we could detect HIV-1 capsids or capsid protein (CA)-containing structures in infected cells following expansion. We infected TZM-bl cells with the non-integration competent HIV-1 NL4-3 derivative, NNHIV, which carries both a deletion in Tat and several point mutations in the integrase [29,30]. At 16 h post-infection, the cells were fixed and subjected to U-ExM and indirect immunostaining using two different antisera against CA [41,77], along with immunostaining against Nup153 and BODIPY staining (Figure 6A,B). To avoid bleedthrough, we used secondary antibodies coupled to spectrally well-separated fluorophores (Alexa 405 Plus and Alexa-647) to visualize the signals of both CA antisera. The robust colocalization of signals of both CA antisera confirmed their specificity and functionality in U-ExM (Figure 6A,B). By focusing on individual CA foci, we identified structures likely representing different stages of the HIV-1 replication cycle. Specifically, on the cytoplasmic side of the nuclear envelope, we detected small, circular CA-containing foci of high intensity in close proximity to Nup153 signals, presumably representing capsids interacting with nuclear pores before transit through the NPC (Figure 6A). The diameter of nuclear envelope-associated CA signals was similar to that of Nup153 foci, indicating they may represent individual viral capsids ca. 100 nm in size. Within the nucleus, we found larger, more diffuse areas of CA signals with a lower degree of colocalization between the two CA antisera (Figure 6B,C). These structures presumably represent accumulations of (possibly disintegrating) viral cores, which have previously been observed within nuclear speckles by electron tomography [30].

To further characterize the nuclear CA-containing structures, we stained samples for membranes using BODIPY, as well as for the nuclear speckle marker SRRM2, the nuclear protein cleavage and polyadenylation-specific factor 6 (CPSF6), and HIV-1 CA. Upon entry into the nucleoplasm, HIV-1 capsids are tightly decorated by CPSF6, which masks CA epitopes in conventional immunostaining [30]. Through indirect immunostaining after U-ExM, we successfully detected both CPSF6 and CA with a high degree of colocalization, appearing as punctate structures presumably representing single viral cores, as well as larger diffuse signals indicative of clusters of (disassembling) capsids (Figure 6B,C). Both structures colocalized with the nuclear speckle marker SRRM2, confirming their identity as nuclear HIV-1 CA-containing structures (and possibly intact viral cores) that have translocated to nuclear speckles (Figure 6C). In one instance, we observed a punctate CA signal close to the nuclear envelope, marked by BODIPY, which presumably represents an HIV-1 capsid that had just traversed the NPC (Figure 6D). Despite its proximity to the nuclear envelope, this CA signal exhibited strong colocalization with CPSF6, consistent with reports of CPSF6 associating with capsids immediately upon nuclear entry [41]. In contrast to CA foci in the inner nucleus, this particular CA focus did not show colocalization with SRRM2 (Figure 6D). These results demonstrate that U-ExM is a valuable tool for identifying and characterizing different stages in the HIV-1 replication cycle.

## 4. Discussion

In this study, we evaluated the usage of the U-ExM protocol to investigate host–virus interactions, focusing specifically on the post-entry stages of the HIV-1 replication cycle. U-ExM allows a four-fold sample expansion in combination with post-expansion labeling, enabling lower relative linkage errors and increased labeling efficiency. While other protocols achieve around 10-fold expansion, they typically require proteolytic digestion, making them incompatible with post-expansion labeling [13,78]. Despite these limitations, higher expansion factors have been achieved by adaptation of gel recipes for a ca. 10-fold expansion [11] or using iterative expansion microscopy for a 14–26-fold expansion [10,12]. However, adapting gel recipes usually requires careful optimization depending on the sample type, complicating their usage [9,11]. Additionally, 10-fold expansion processes that employ alternative gel chemistries often necessitate specialized equipment to eliminate oxygen during the gel polymerization process [78]. Notably, very recent results show promising advances for the development of novel one-step gelation strategies that allow robust expansion up to 12-fold, which can be used to probe chromatin architecture, NPC structure, as well as virion morphology in 3D [79]. However, the wider feasibility of these approaches remains to be seen. In iterative expansion microscopy, gels are re-embedded into a secondary gel following an initial expansion step, enabling a subsequent round of expansion [10,12]. While these approaches enable visualization of molecular details at a resolution below 20 nm, they are considerably more time-consuming and complex than one-step expansion protocols. In addition, a second expansion may compromise the robustness of the protocol, leading to greater variability in final expansion compared to one-step expansion [12].

By integrating U-ExM with Airyscan microscopy, which delivers improved resolution while being more user-friendly than STED or SMLM [20], we successfully resolved the subunit arrangement of structures with diameters under 150 nm, despite the usage of indirect immunofluorescence staining (Figure 5A,B). Although U-ExM inherently results in low relative linkage errors when using primary and secondary antibodies, localization precision can be further improved by the use of self-labeling tags, such as the SNAP-tag [75] (Figure 5C–F and Appendix A). Our findings demonstrate that by employing biotin-coupled reporters, we can effectively detect self-labeling protein tags after expansion. This strategy has been successfully applied to various targets [80,81] and can be adapted for the detection of any biomolecule tagged with biotin.

Although biotin labeling represents an approach that minimizes linkage errors, it may require cell permeabilization to introduce biotin-coupled moieties and appears to be more prone to background signal compared to traditional immunostaining (Figure 5C–F and Appendix A). This increased background is likely due to the presence of endogenous biotinylated molecules. Consequently, antibody labeling remains the most accessible avenue for target detection in U-ExM. In our study, we evaluated a panel of antibodies against viral and cellular targets for use in U-ExM and found that most were compatible with post-expansion staining, consistent with previous reports [36]. However, we faced challenges in establishing reliable conditions for antibody staining of GFP tags and lamin, despite other studies successfully visualizing GFP after U-ExM [36,82].

We observed that the nuclear compartment was well preserved after U-ExM, while the cytoplasmic compartment needed optimization of fixation for adequate sample preservation, a phenomenon that has been well documented [50]. While cryofixation effectively preserves sample integrity, it requires specialized equipment and prolongs the U-ExM sample preparation process by an additional day [50]. Additionally, previous studies indicate that cryofixation may need further optimization to enhance antibody-mediated detection of NPC structures in expansion microscopy [12]. We found that substituting PBS with different buffer solutions optimized for sample preservation markedly improved post-expansion results, achieving outcomes comparable to those with cryofixation (Figure 2A and Appendix A).

Using U-ExM, we probed the HIV-1 nuclear replication cycle and readily detected HIV-1 viral cores in different stages of the post-entry process (Figure 6). Within the nucleus, individual particles exhibited strong colocalization with the nuclear protein CPSF6, even when positioned adjacent to the nuclear envelope, suggesting they had just entered the nucleus (Figure 6C,D). This finding is consistent with reports showing that capsids are immediately decorated with CPSF6 upon arrival in the nucleoplasm [41,83]. Additionally, we identified more diffuse CA signal areas within the nucleus that colocalized with the nuclear speckle marker SRRM2, presumably representing accumulated and possibly degrading viral cores (Figure 6B,C). These observations support other reports indicating that multiple, apparently morphologically intact capsids gather within nuclear speckles prior to viral genome uncoating and integration into speckle-associated domains [30,31,32,34].

Of note, we could readily observe both CA and CPSF6 signals in regions that presumably represented intact viral cores (Figure 6C,D). Previously, due to the masking of CA epitopes by the CPSF6 coat, simultaneous detection of CA and CPSF6 was often only possible in classical immunofluorescence after chemical extraction [30]. Here, however, both targets were accessible in parallel with high fidelity, likely due to the exposure of epitopes during expansion. To our knowledge, this is the first clear demonstration of epitope unmasking during U-ExM. This decrowding of dense environments is especially valuable, as it not only enables investigation of viral structures at higher resolution but also unlocks new possibilities for the detection of features that have previously been masked.

Due to their size, singular capsids cannot be faithfully resolved in conventional fluorescence microscopy. At the same time, methods that allow visualization of capsid with appropriate resolution, such as STED and SMLM, are technically challenging and time-consuming—especially when scanning larger volumes—which particularly complicates detection of rare events. In this context, we demonstrate that using U-ExM in combination with Airyscan microscopy allows for clear differentiation of individual capsids as punctate CA signals (Figure 6). This combination facilitates targeted investigations of virus–host interactions at the nanoscale. At the same time, Airyscan microscopy of expanded samples allows fast image acquisition and scanning of large 3D volumes. By combining U-ExM with Airyscan microscopy or other microscopy techniques that achieve similar resolution [84,85], we can identify rare events with high resolution, opening up new avenues for detection of specific host–virus interactions at super-resolution.

## 5. Conclusions

Here, using the visualization of HIV-1 post-entry events as an example, we illustrate that expansion microscopy is a promising method for the investigation of virus-cell interactions, especially within the nuclear compartment. U-ExM is a robust and versatile protocol applicable to various cell types and structures, effectively bridging the gap between fluorescence and electron microscopy while enhancing the resolution without the need for specialized equipment. As with any super-resolution technique, labeling becomes critical in expansion microscopy due to the linkage errors conferred by different molecular visualization tools. Although post-expansion labeling protocols like U-ExM improve localization accuracy, the impact of antibody size on linkage error remains significant. Thus, we recommend adopting more direct labeling methods, such as self-labeling tags, moving forward [75]. Additionally, alternative labeling strategies that minimize linkage errors, such as directly labeled camelid antibodies [86] or genetic code expansion with click labeling [87], could enhance the localization accuracy in expansion microscopy. By further increasing the expansion factor and integrating super-resolution imaging methods such as Airyscan microscopy, structured illumination microscopy (SIM), and STED, we can explore virus–host interactions at unprecedented scales.

## Figures and Tables

**Figure 1 viruses-16-01610-f001:**
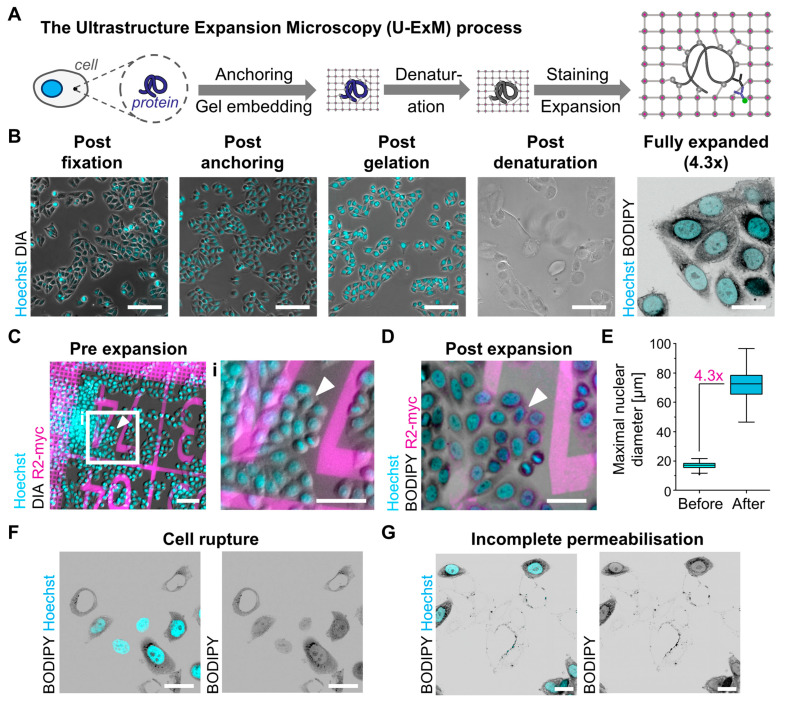
U-ExM enables robust 4.3-fold expansion with minor macroscopic distortions. (**A**) A schematic overview illustrating the different steps involved in the U-ExM procedure. (**B**) TZM-bl cells, fixed with PFA/GA, stained for chromatin (Hoechst, cyan) and membranes (BODIPY, right panel, gray) and imaged at different steps of the U-ExM process. Scale bar: 100 µm. (**C**) TZM-bl cells seeded onto an 18 mm GelMap slide patterned with R2-myc-his-ATTO 647N (magenta), fixed with PFA/GA, and stained for chromatin (Hoechst, cyan). Scale bar overview, 100 µm; scale bar enlargement, 50 µm. Arrowhead marks the position of the same cluster of cells before and after expansion in (**C**,**D**). (**D**) TZM-bl cells seeded onto an 18 mm GelMap slide, fixed with PFA/GA, expanded by U-ExM, and stained for chromatin (Hoechst, cyan), membranes (BODIPY, gray), and R2-myc (magenta). Scale bar, 200 µm. (**E**) Measurement of the Feret diameter of the nuclei of the same set of TZM-bl cells seeded onto an 18 mm GelMap slide, fixed with PFA/GA and stained for chromatin, before and after expansion; *n* = 168; median_before_ = 17.03 µm, median_after_ = 72.63 µm; data plotted according to Tukey box plot. (**F**,**G**) TZM-bl cells stained for chromatin (Hoechst, cyan) and membranes (BODIPY, gray) exhibiting expansion artifacts. Scale bar, 100 µm. All scale bars in expanded samples reflect post-expansion sizes.

**Figure 2 viruses-16-01610-f002:**
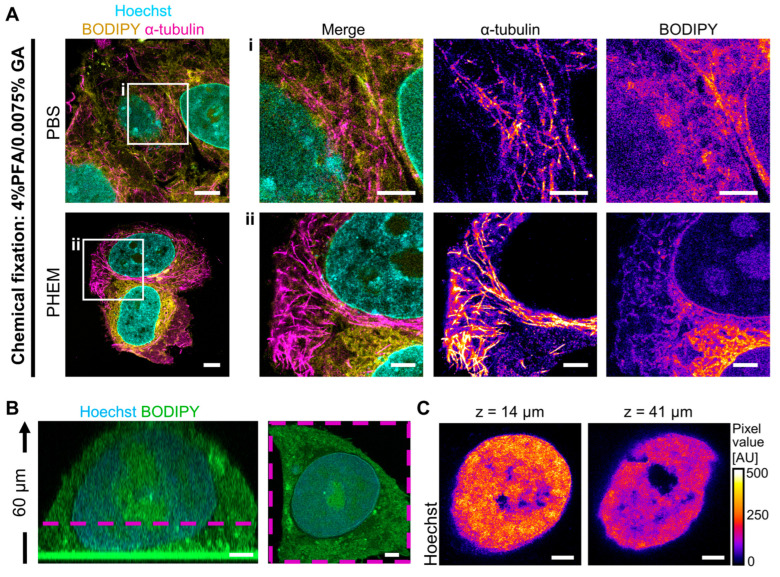
Sample preparation and imaging optimization are critical for expanded samples. (**A**) TZM-bl cells fixed with PFA/GA in either PBS or PHEM buffer, expanded and stained for chromatin (Hoechst, cyan), membranes (BODIPY, yellow), and α-tubulin (magenta). Scale bar overview, 20 µm; scale bar enlargements, 10 µm. (**B**) 3D structure of an expanded TZM-bl cell, stained for chromatin (Hoechst, cyan) and membranes (BODIPY, green). First panel, zy-slice; second panel, xy-slice. The magenta dashed line denotes the position of the xy slice in z. Scale bar, 10 µm. (**C**) Single slices of the same expanded TZM-bl cell as in (**B**), stained for chromatin (Hoechst). Scale bar, 10 µm. All scale bars in expanded samples reflect post-expansion sizes.

**Figure 3 viruses-16-01610-f003:**
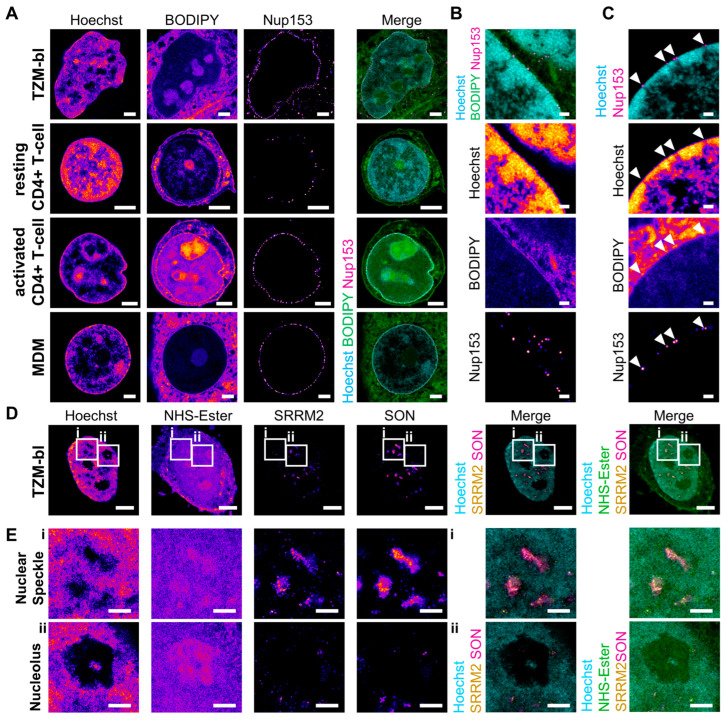
U-ExM allows expansion of various cell types and cellular structures. (**A**) TZM-bl cells, resting and activated CD4+ T-cells, and MDM fixed with PFA/GA, expanded and stained for chromatin (Hoechst, cyan), membranes (BODIPY, green), and Nup153 (magenta). Scale bar, 10 µm. (**B**) Cropped region of the nucleus of an MDM fixed with PFA/GA, expanded and stained for chromatin (Hoechst, cyan), membranes (BODIPY, green), and Nup153 (magenta) showing double membrane layers of the nuclear envelope in membrane staining. Scale bar, 2 µm. (**C**) Cropped region of the nucleus of an MDM fixed with PFA/GA, expanded and stained for chromatin (Hoechst, cyan), membranes (BODIPY, not shown in merge), and Nup153 (magenta), showing heterochromatin-free channels beneath the NPCs in the nucleus. Scale bar, 2 µm; arrowheads mark the position of NPC. (**D**) A TZM-bl cell fixed with PFA/GA, expanded and stained for chromatin (Hoechst, cyan), protein density (NHS-ester, green), SRRM2 (yellow), and SON (magenta), showcasing different intranuclear compartments. Scale bar, 10 µm. (**E**) Enlargements from (**D**) highlighting nuclear speckles and nucleolus. Scale bar, 5 µm. All scale bars in expanded samples reflect post-expansion sizes.

**Figure 4 viruses-16-01610-f004:**
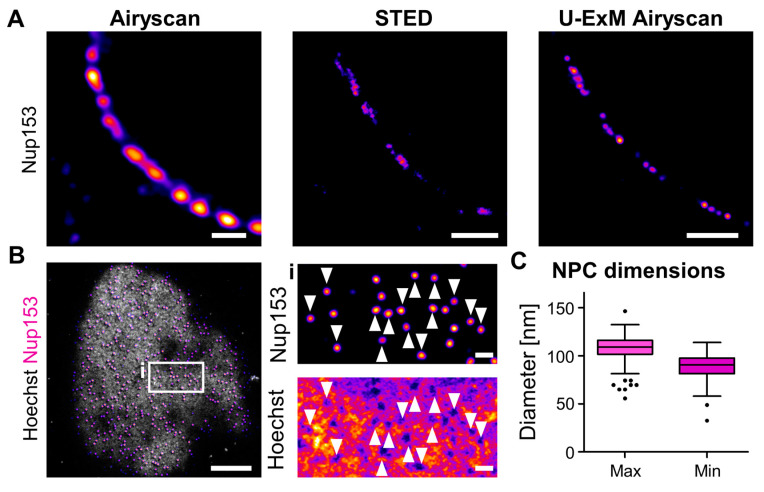
NPC diameter is well conserved after U-ExM. (**A**) Different MDM fixed with PFA/GA and either stained for Nup153 and imaged with Airyscan microscopy or STED microscopy or expanded, stained for Nup153 and imaged with Airyscan microscopy. Scale bar of Airyscan and STED panel, 1 µm; scale bar U-ExM panel, 4 µm. (**B**) Maximum intensity projection (MIP) of the bottom of an expanded U2OS cell nucleus, stained for chromatin (Hoechst, gray) and Nup153 (magenta). Scale bar, 10 µm; scale bar enlargement, 1 µm. Arrowheads mark the position of select NPC. (**C**) Analysis of NUP153 signal to determine NPC dimension, measured as maximum and minimum Feret diameters. Sample size (*n* = 222); median_max diameter_ = 109 nm, median_min diameter_ = 91 nm; data blotted according to Tukey box plot with outliers shown as single data points. All scale bars in expanded samples reflect post-expansion sizes.

**Figure 5 viruses-16-01610-f005:**
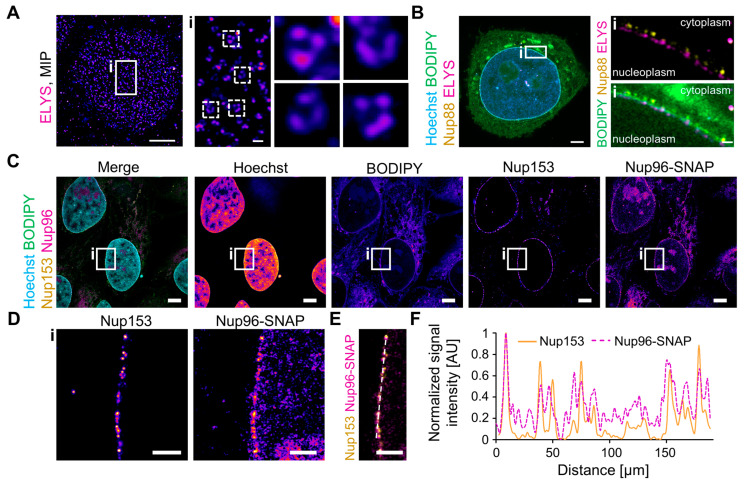
U-ExM allows visualization of nanoscopic structures. (**A**) MIP of a TZM-bl cell fixed with PFA/GA, expanded and stained for ELYS (magenta), showing ring-like structures (enlargements). Scale bar overview, 10 µm; scale bar enlargement, 1 µm. (**B**) Single z-slice of a TZM-bl cell fixed with PFA/GA, expanded and stained for chromatin (Hoechst, cyan), membranes (BODIPY, green), Nup88 (yellow), and ELYS (magenta). Scale bar overview, 10 µm; scale bar enlargement, 1 µm. (**C**) U2OS cells stably expressing Nup96-SNAP fixed with PFA/GA, permeabilized with saponine, and stained with BG-Biotin before expansion and staining for chromatin (Hoechst, cyan), membranes (BODIPY, green), Nup153 (yellow), and biotin-labeled Nup96 via fluorescently conjugated Streptavidin (magenta). Scale bar, 20 µm. (**D**) Enlargement from (**C**) showing a portion of the nuclear envelope. Scale bar, 5 µm. (**E**) Composite image of the region shown in (**D**). Scale bar: 5 µm; line denotes position of line profile for (**F**) normalized signal intensity of the Nup153 (yellow) and Nup96-SNAP (magenta, dotted line) signal over the line shown in (**E**). Line profiles were aligned to the maximum peak. All scale bars in expanded samples reflect post-expansion sizes.

**Figure 6 viruses-16-01610-f006:**
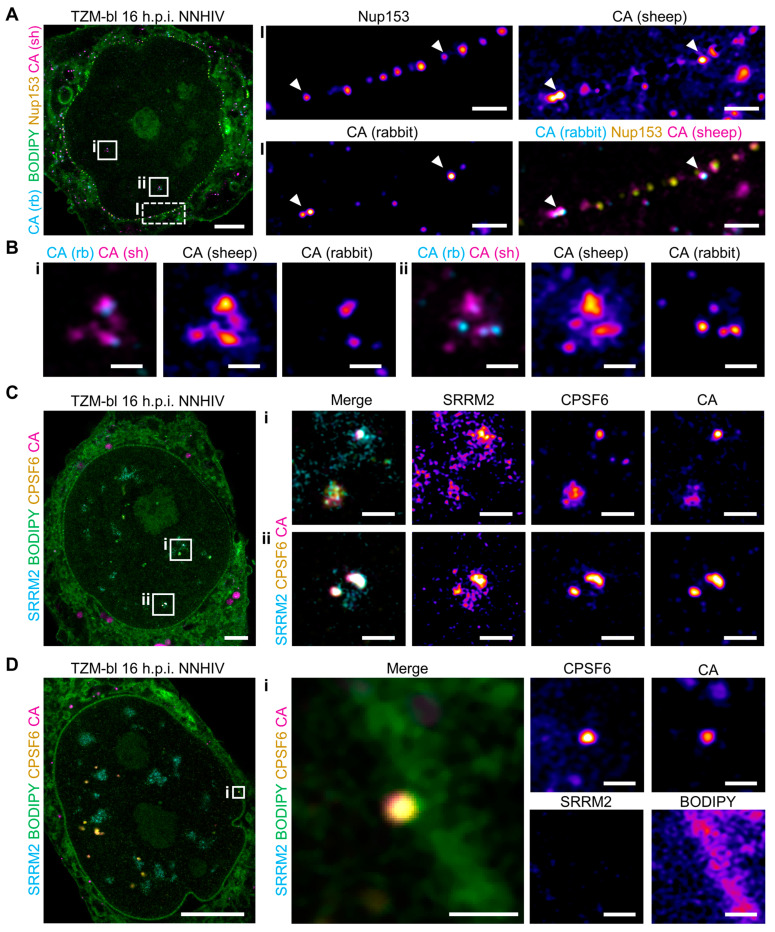
U-ExM can visualize HIV-1 CA-containing structures upon nuclear entry and within the nucleus. (**A**) A TZM-bl cell infected with NNHIV, fixed with PFA/GA at 16 h.p.i., expanded and stained for CA with two different antisera (rabbit α-CA, cyan; sheep α-CA, magenta) as well as Nup153 (yellow) and membranes (BODIPY, green). The sheep α-CA channel was corrected for bleedthrough from the Nup153 channel. Scale bar overview, 20 µm. Enlargements show HIV-1 CAs (cyan/magenta) interacting with NPC complexes (yellow), marked by arrowheads. Scale bar enlargements, 2 µm. (**B**) Enlargements from (**A**) showing intranuclear CA signals (rabbit α-CA, cyan; sheep α-CA, magenta). Scale bar, 1 µm. (**C**,**D**) TZM-bl cells, infected with NNHIV, fixed with PFA/GA at 16 h.p.i., expanded and stained for the nuclear speckle marker SRRM2 (cyan), membranes (BODIPY, green), CPSF6 (yellow), and CA (sheep α-CA, magenta). Scale bar overview, 20 µm. (**C**) Enlargements show intranuclear CA signals (magenta) showing colocalization with SRRM2 (cyan) and CPSF6 (yellow). The CA channel and the SRRM2 channel were corrected for bleedthrough from the BODIPY channel. Scale bar enlargements, 2 µm. (**D**) Enlargement shows intranuclear CA signals (magenta) with CPSF6 (yellow) close to nuclear envelope marked with BODIPY (green). Scale bar enlargements, 1 µm. All scale bars in expanded samples reflect post-expansion sizes.

**Table 1 viruses-16-01610-t001:** Different steps of the U-ExM protocol and their approximate duration.

	Step	Time
Day 1	Anchoring	3.5 h
	Gelation	1.5 h
	Denaturation	2 h
	First expansion	1.5 h
	PBS shrinking	0.5 h
	Blocking	0.5 h
	Primary antibody incubation	overnight
Day 2	Washing	1 h
	Secondary antibody incubation	2.5 h
	Washing	1 h
	[NHS-ester staining]	[1.5 h]
	[Washing]	[0.5 h]
	Second expansion and BODIPY staining	overnight
Day 3	Final expansion steps	1 h

**Table 2 viruses-16-01610-t002:** Different fixation conditions for optimization of sample preservation.

Fixation Condition	Buffer Solution
Cryofixation	Medium, pH 7.4 Hepes-buffered Hanks’ balanced salt solution (HHBSS)/10% glycerol, pH 7.4
Chemical fixation (4% PFA/0.0075% GA)	PBS buffer, pH 7.4 PHEM buffer, pH 7.4
	PEM buffer, pH 6.8
(0.25% Triton X-100/ 0.3% GA; 2% GA)	Cytoskeleton buffer, pH 6.1

## Data Availability

The data presented in this study are available upon request from the corresponding author.

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
