# Peer review of "Expanding Insights: Harnessing Expansion Microscopy for Super-Resolution Analysis of HIV-1–Cell Interactions"

_viruses, 2024, doi:10.3390/v16101610_

Round 1

Reviewer 1 Report

Comments and Suggestions for Authors

This manuscript by Petrich and colleagues is a proof of principle study examining the utility of expansion microscopy to visualize late steps of nuclear HIV-1 infection. In this regard, it is timely and well written. In particular, the authors do a good job in explaining the approach and limitations of different expansion microscopy approaches (ie staining before or after expansion) and the materials and methods section is detailed and clear to any microscopist.

In the results presented, the authors commendably characterize a number of abnormalities which occur during expansion, which was appreciated by this reviewer. Rather than simply jumping to cells that have successfully expanded, the highlighting of this issues will likely be of great utility to researchers looking to replicate these techniques in their own studies. Similarly, they rigorously evaluate the effect of  different fixation approaches and the effect of these approaches on downstream expansion processing

One of the more provocative observations is the reduction in chromatin in the nucleus proximal to NPCs. The authors don’t comment much on this. Is this seen in non-expanded cells using super resolution approaches/is this documented in the literature. Do the authors see this regularly? 

In the aggregate, this is a very technically strong paper. I can imagine some level of editorial question regarding the lack of a bonafied “discovery” related to the HIV-1 lifecycle, but I would argue that the publication of a very comprehensive study such as this, evaluating the degree to which different aspects of the protocol affect the final result, is and will be valuable to the field. The alternative is trying to cram all of this information into another paper on a different topic, and in that case, the methodology would surely not be described as well as in this paper, leaving researchers to learn many of the lessons in this paper themselves when they try to use this approach. 

Author Response

Comment 1: One of the more provocative observations is the reduction in chromatin in the nucleus proximal to NPCs. The authors don’t comment much on this. Is this seen in non-expanded cells using super resolution approaches/is this documented in the literature. Do the authors see this regularly?

Response 1: We thank the reviewer for their positive feedback and appreciate their interest in the reduced chromatin density proximal to nuclear pore complexes (NPCs). Indeed, we see this quite regularly in cells that we have expanded, including also U2OS and T-cells. This phenomenon has been observed in multiple studies, including in studies carried out in non-expanded cells (typically using electron microscopy). To emphasize that this is a known feature of chromatin architecture, we have amended the relevant section in the revised text (lines 596-598).

Reviewer 2 Report

Comments and Suggestions for Authors

In this article by Petrich et al. there is the description of a pipeline for 4x ExM aimed to study HIV-1 cell interactions. The text is clear and the science is appropriate for Viruses. The usage of ExM into virology research, namely HIV, has great potential and has been mostly under-explored. Most of the conclusions and discussion are supported by the data with on major technical concern on data analysis that should be addressed for rigour purposes, as well a clear detailing how the data to be shown was selected, and describing so in the main text (as there is no analysis of randomized data).

Major Issues:

Main issue: The authors use segmentations to calculate sizes of structures through Feret diameters. This method is highly prone to errors due to the processing done for the segmentation and the segmentation itself. Please provide measurements (in parallel) for full width at half maximum (FWHM) measurements in traced lines beyond the analyzed structures (that can expand from the Feret lines, for example) showing the diameter and the resolution. Also, fiducial measurements to establish the limits of the resolution of the imaging (under ExM) should be used. A possible control: microtubules (that are ~25nm, thus 25x4=100nm would be under the difraction limit and would provide the resolution of the imaging performed).

- The methods of data analysis of the data are over processed in some cases. The manuscript uses mean filters and erosion and localized bleedthrough corrections (the method likely results in lower intensity measurements in the areas with bleedthrough compared to areas without, as it's only corrected in that area). Since there are figures mentioning intensity of viral structures it might be technically more sound to apply an overall subtraction to draw such conclusions.

- The methods don't describe how to generate macrophages from monocytes (no cytokines mentioned). The method as it written would leave the PBMCs as collected (mostly monocytes stuck on the surface).

-For rigour purposes, please clearly detail how the specific images were selected (or ignored/discarded) as there is only a selection of data without quantification of the totality of the data (viral particles for example). 

Minor issues:

- Figures could use clarification if the scale bars were adjusted for the expansion or are 'raw'. It could be helpful for a reader to always show graphically when an image is from an ExM experiment. 

- Please discuss why the used specific methods compare to others (not based on denaturation) for fluorescence loss during U-ExM. Under the literature and my experience, it was not the case. The (data not provided) should be provided and quantified.

- Please discuss how heat denaturation allows better staining post-expansion, could it generally mean that the epitopes themselves are not "expanded"?

-(73-77) The authors describe the limitation of antibody staining (30nm) when using primary and secondary antibodies. It is possible to use directly conjugated antibodies as they are easily commercial available in the presence. Nanobodies would also reduce these limitations (but in this case the authors describe so in the discussion section).

- (105) The word genome uncoating should be revised. As it is not common in the field (or clear).

- The authors should use CA, when referring to structures with CA, and capsid to the viral structure for clarity purposes. Please consider using viral cores or RTC throughout the text (for example lines 104-105).

- Line 158- NNHIV is used before its definition in line 162.

- Consider including the working distances of the used objectives. A challenge for ExM is to provide correct imaging at high magnification when Z expands.

- In line 802 there's mention of nuclear CPSF6, are there any CPSF6+ particles in the cytoplasm, for example in Fig6D there a particle on the top left.

Reviewer 3 Report

Comments and Suggestions for Authors

Ultrastructure expansion microscopy (U-ExM) is a relatively novel approach that has not been widely used to study virus-host interactions. In their manuscript, Petrich et al describe how this approach can be employed for such studies, providing a detailed protocol, characterising the expansion they achive to ensure that the ultrastructure of the cell is preserved (mainly focusing in the nucleus), and use this approach to gain some insights into the different HIV-1 capsid assemblies present in the nucleus.

While the manuscript only provides minor advances in HIV research, and it is mostly a methodological description of U-ExM, it is very well written, and the detailed insights of it provides of U-ExM will make it a valuable contribution to the virology field.

Overall there are no major issues I can find in the manuscript, but I think authors should consider the following minor points:

-              Methods, section 2.5.2: Bleedthrough correction. I don’t think this is a common approach, and reading the authors’ description is unclear if this was done only for BODIPY or for other labels. I would argue that the most straightforward approach is to use spectrally well-separated fluorophores, as the authors do for the CA antisera. Authors should be clearer in the instances they performed bleedthrough correction, and provide some references supporting the approach they followed.

-              Line 648: referring to ELYS, the authors say “we successfully visualized ring-like structures”. The authors should acknowledge that most of the ELYS signal is not forming associated to rings, according to the enlargement from Fig 5A.

-              Lines 672-774: “specific Nup96-SNAP signals colocalized well with the Nup153 signal”. How did the authors distinguish between “specific” and “unspecific” Nup153 signal? Is this based on membrane-association?

-              Figs 6A and B: the CA sheep antiserum seems to be much more unspecific than the CA rabbit antiserum, and it is not clear to me why the authors used two antisera. Authors should either explain why they used the two different antisera, or remove the results for the CA sheep antiserum.

-              Lines 728-729: “this particular capsid did not show colocalization with SRRM2 (Figure 6D)”. Authors should show the SRRM2 signal for this capsid, to support this statement.

Author Response

We thank the reviewer for their positive comments and valuable feedback to improve the clarity of our manuscript. 

Comment 1: Methods, section 2.5.2: Bleedthrough correction. I don’t think this is a common approach, and reading the authors’ description is unclear if this was done only for BODIPY or for other labels. I would argue that the most straightforward approach is to use spectrally well-separated fluorophores, as the authors do for the CA antisera. Authors should be clearer in the instances they performed bleedthrough correction, and provide some references supporting the approach they followed.

Response 1: The utilized bleedthrough correction approach has been well-characterized for scenarios where spectral bleedthrough occurs in one direction (Zimmermann 2005). We have also addressed the theory behind our bleedthrough correction approach in more detail in the response to reviewer 2. To improve understandability and add context for this method, we have expanded the relevant materials and methods section and provide additional supplementary figures illustrating the bleedthrough correction process as well as comparisons of images before and after processing (see new supplementary figure 1 and lines 343-362).

We agree that usage of spectrally well-separated fluorophores represents the ideal avenue for limiting crosstalk. Nevertheless, bleedthrough correction was only necessary for images shown in Fig 6, where Nup153-StarOrange showed bleedthrough into the CA-StarRed channel in panel A, and for images in panel B, where BODIPY-TR showed bleedthrough into the CA-StarRed and the SRRM2-AF405 channels due to spectral overlap and weak overall fluorescence in the CA and SRRM2 channels. To enhance clarity, we have revised our figure legends to clearly mark those images and channels which have been processed (see Figure 6, lines 774-783).

Zimmermann T. Spectral imaging and linear unmixing in light microscopy. Adv Biochem Eng Biotechnol. 2005;95:245-65. doi: 10.1007/b102216.

Comment 2: Line 648: referring to ELYS, the authors say “we successfully visualized ring-like structures”. The authors should acknowledge that most of the ELYS signal is not forming associated to rings, according to the enlargement from Fig 5A.

Response 2: We thank the reviewer for pointing this out and have changed the text to reflect these observations (lines 682-685).

Comment 3: Lines 672-774: “specific Nup96-SNAP signals colocalized well with the Nup153 signal”. How did the authors distinguish between “specific” and “unspecific” Nup153 signal? Is this based on membrane-association?

Response 3: Indeed, we classified “specific” Nup96-SNAP (and Nup153) as associating with the nuclear envelope as marked by the BODIPY signal. We thank the reviewer for highlighting that this distinction requires clarification and have changed the text accordingly (lines 711-712).

Comment 4: Figs 6A and B: the CA sheep antiserum seems to be much more unspecific than the CA rabbit antiserum, and it is not clear to me why the authors used two antisera. Authors should either explain why they used the two different antisera, or remove the results for the CA sheep antiserum.

Response 4: In Figure 6A and B our decision to use two well-characterized antisera (Bejarano et al., 2019, Hanne et al., 2016) against the same target (i.e. CA), resulted from the observation that structures stained with either antiserum were usually small, numerous, and resembled unspecific staining. To ensure that both antisera indeed detected true CA-containing structures, we employed them simultaneously to probe if the detected structures were colocalizing, and therefore, indicating specificity. We agree that the sheep antiserum has higher background levels than the rabbit antiserum. However, after we had confirmed that both sera detected the same structures, we decided to use the sheep antiserum in the following experiments as we aimed for simultaneous detection of CA with two other HIV-1 relevant targets (namely CPSF6 and SRRM2) using known and well-characterized antibodies produced in rabbit and mouse, respectively. This required the use of an antibody or antiserum produced in another species. Indeed, as most commercial antibodies are derived from mice, rabbits or sometimes rats, we consciously decided on characterizing and utilizing the sheep antiserum, as it provided more flexibility in detecting multiple targets at once. We have amended the text to explain our reasoning more clearly and hope this additional clarification provides some context over our decision to utilize the respective antiserum (lines 734-741).

  1. A. Bejarano et al., “HIV-1 nuclear import in macrophages is regulated by CPSF6-capsid interactions at the nuclear pore 973 complex,” Elife, vol. 8, 2019, doi: 10.7554/ELIFE.41800.
  2. Hanne et al., “Stimulated Emission Depletion Nanoscopy Reveals Time-Course of Human Immunodeficiency Virus Proteo-1052 lytic Maturation,” ACS Nano, vol. 10, no. 9, pp. 8215–8222, Sep. 2016, doi: 10.1021/ACSNANO.6B03850/ASSET/IMAGES/NN-1053 2016-03850E_M003.GIF.

Comment 5: Lines 728-729: “this particular capsid did not show colocalization with SRRM2 (Figure 6D)”. Authors should show the SRRM2 signal for this capsid, to support this statement.

Response 5: We thank the reviewer for this very valid observation and have adapted the figure to show the SRRM2 channel as well (see Figure 6D).